

# Effects of Water-soluble Organic Carbon on Aerosol pH

Michael A. Battaglia, Jr.[1], Rodney J. Weber[2], Athanasios Nenes[2,3,4,5], Christopher J. Hennigan[1*]

[1] Department of Chemical, Biochemical and Environmental Engineering, University of Maryland, Baltimore County, Baltimore, MD 21250, USA

[2] School of Earth and Atmospheric Sciences, Georgia Institute of Technology, Atlanta, GA 30332, USA

[3] Institute for Chemical Engineering Sciences, Foundation for Research and Technology – Hellas, Patras, 26504, Greece

[4] Institute for Environmental Research and Sustainable Development, National Observatory of
Athens, Palea Penteli, Athens, 15236, Greece

[5] Laboratory of Atmospheric Processes and their Impacts, School of Architecture, Civil and Environmental Engineering, Ecole Polytechnique Fédérale de Lausanne, CH-1015, Lausanne, Switzerland

[*]To whom correspondence should be addressed: email hennigan@umbc.edu; phone: (410) 455-3515

## Abstract

Water soluble organic carbon (WSOC) is a ubiquitous and significant fraction of fine particulate matter. Despite advances in aerosol thermodynamic equilibrium models, there is limited

understanding on the comprehensive impacts of WSOC on aerosol acidity (pH). We address this limitation by studying submicron aerosol that represent the two extremes in acidity levels found in the atmosphere: strongly acidic aerosol from Baltimore, MD, and weakly acidic conditions characteristic of Beijing, China. These cases are then used to construct mixed inorganic/organic single-phase aqueous particles, and thermodynamically analyzed by the E-AIM and

ISORROPIA models (in combination with activity coefficient model AIOMFAC) to evaluate the effects of WSOC on the H$^+$ ion activity coefficients ($\gamma_{H+}$) and activity (pH). We find that addition of organic acids and non-acid organic species concurrently increases $\gamma_{H+}$ and aerosol liquid water. When allowed to modulate pH, these effects mostly offset each other, giving pH changes of < 0.6 pH units even if organics dominate aerosol dry mass (in excess of 60%).

Surprisingly, non-acidic WSOC compounds were found to have a larger effect on pH than



organic acids owing to their stronger impacts on $\gamma_{H+}$. The model simulations were run at 70%, 80%, and 90% relative humidity (RH) levels and the effect of WSOC was inversely related to RH. At 90% RH, WSOC altered aerosol pH by up to ~0.2 pH units, though the effect was up to ~0.6 pH units at 70% RH. The offsetting nature of these effects suggests that aerosol pH is

sufficiently constrained by the inorganic constituents alone and under conditions where liquid-liquid phase separation is not anticipated to occur.

## 1. Introduction

The acidity of atmospheric particles is thought to play a critical role in many physicochemical processes. Some of these processes include sulfur oxidation and halogen chemistry, with

important implications for the formation of sulfates (Chameides, 1984) ; the oxidation of volatile organic compounds (VOCs), and ozone formation in marine environments (Keene et al., 1998); the gas-particle partitioning of many semi-volatile species (Ahrens et al., 2012; Keene et al., 2004); and, enhancements to secondary organic aerosol (SOA) formation (Hallquist et al., 2009). The inorganic salt constituents in atmospheric particles, such as ammonium sulfate ($(NH_4)_2SO_4$)

and ammonium bisulfate ($NH_4HSO_4$), contribute to particle acidity and water content, with effects on aerosol radiative forcing (Seinfeld and Pandis, 2016). In addition to the physicochemical effects within particles, their bulk acidity can affect health, both of environmental ecosystems and the human populations therein either directly (Gwynn et al., 2000; Peters et al., 1996; Schindler, 1988; Spengler et al., 1996; Fang et al., 2017; Johnson et al.,

2008), or by their effects on nutrient deposition (Myriokefalitakis et al., 2016; Myriokefalitakis et al., 2018; Kanakidou et al., 2016; Nenes et al., 2011).

pH, the parameter serving to define and describe the acidity of aqueous solutions, often has no direct correlation with proxy measurement methods such as aerosol strong acidity or aerosol total acidity (Hennigan et al., 2015; Song et al., 2018). The most accurate predictions of aerosol pH



come from aerosol thermodynamic equilibrium models constrained by both aerosol and gas-phase measurements (i.e., "forward mode" calculations), or from the measured gas-particle partitioning of semi-volatile species, including ammonia, nitric acid, or oxalic acid, which provide direct insight to the pH (Hennigan et al., 2015). Both approaches utilize aerosol and gas-

phase composition measurements, along with the temperature and relative humidity, to obtain the pH of an aerosol distribution. Consistent pH values are obtained when the assumptions about aerosol mixing and equilibrium are met (Guo et al., 2018a; Guo et al., 2018b).

Different aerosol thermodynamic equilibrium models have been developed through the years, each with their unique sets of assumptions, simplifications and approach to obtain the

composition at thermodynamic equilibrium. The Extended Aerosol Inorganics Model (E-AIM, http://www.aim.env.uea.ac.uk/aim/aim.php) and the ISORROPIA-II model (http://isorropia.eas.gatech.edu) are widely used to calculate aerosol pH for atmospheric and experimental particle distributions (Guo et al., 2017; Guo et al., 2016; Guo et al., 2015; Wang et al., 2016). The AIOMFAC model (http://www.aiomfac.caltech.edu/model.html) offers the most

extensive treatment of organic-inorganic interactions (Ganbavale et al., 2015) of models to date, but is primarily an activity coefficient model that does not solve full thermodynamic equilibrium calculations or phase diagrams as E-AIM and ISORROPIA do. At present, E-AIM, ISORROPIA, and AIOMFAC are widely used for atmospheric applications due to their demonstrated predictive capabilities and their accessibility: they are freely available online, and

include resources and user guides to facilitate their application and use.

One key difference among the models is their treatment of organics. ISORROPIA does not include organic species. E-AIM functions similarly to ISORROPIA when considering inorganic species, but in addition offers a limited library of organic acids (included by UNIFAC methods





or fitted activity equations). AIOMFAC offers wide support for organic components, but does not solve the thermodynamic equilibrium problem for system inputs. These models, and most others, do not treat organics in a way that is comprehensive (that is, simultaneous consideration of activity coefficient calculations and thermodynamic equilibrium calculations). However,

these simplified thermodynamic models do seem to capture the partitioning of inorganic species well, even when organic components are present in large quantities, which indicates that pH should be captured well (Guo et al., 2018a).

The effects of WSOC on aerosol pH come through two primary means: dilution of the aqueous phase by aerosol liquid water associated with the organic fraction ($W_o$); and changes to the

hydrogen ion activity coefficient and thus hydrogen activity in solution ($\gamma_{H+}$ and $a_{H+}$, respectively). The total contribution of organics to aerosol water can be as much as 30-50% of total fine particle aerosol water in the polluted Beijing winter haze events (Tan et al., 2018; Huang et al., 2014), 40-50% in the southeast United States (Nah et al., 2018; Guo et al., 2015), and the eastern Mediterranean (Bougiatioti et al., 2016). The effects of organics (soluble and

insoluble) on aerosol pH under conditions of LLPS are more complex. Free $H^+$ ion is predicted to have increased association with $SO_4^{2-}$ to form $HSO_4^-$ when organic compounds are in the same phase as inorganic ions, resulting in a 0.1 pH unit increase in aerosol pH (Pye et al., 2018). The isolation of the organic components in a separate phase (LLPS condition) has also demonstrated an effect on aerosol pH by altering the $NH_3$ partitioning behavior, and thus the ability of this

basic gas to neutralize aerosol acidity. The inclusion, or lack thereof, of organic compounds was predicted to have a greater effect on $NH_3$ partitioning behavior than the inclusion, or lack thereof, of nonvolatile cations, nitrate, and chloride (Guo et al., 2018a). In addition to these effects, AIOMFAC predicts that any organic presence in the same phase with inorganic



constituents drives free $H^+$ to increased association with sulfate to form bisulfate, a compound predicted to be more miscible with organics than $H^+$ and small cations. AIOMFAC was used to show the organic phase still contains a significant amount of inorganics, affecting the partitioning medium by inclusion of the inorganic ions and their associated water, lowering the

mole fraction and activity of organics, and increasing particle phase partitioning of organic compounds with O:C > 0.6 (Pye et al., 2018). In the case where multiple phases do exist, there is anticipated to be a primary-organic (PO) and primarily-inorganic (PI) phase, each of which contains $H^+$ in equilibrium with the other phase. In the case where the vast majority of inorganics are partitioned to the PI phase, the pH is not anticipated to change drastically, as $H^+$ is

also required to be in equilibrium with the gas phase, which may explain the results of (Pye et al., 2018). In the case of phase separation where the PO phase contains considerable amounts of inorganic species, there exists the possibility of a PI phase with substantially-altered $H^+$ activity, and therefore, the potential for substantially-altered aerosol pH.

Aerosol pH can also be directly affected by the uptake of organic acid species, whose

dissociation produces free hydronium ions in the particle aqueous phase. Carboxylic acids represent a highly abundant moiety in atmospheric OA (Yatavelli et al., 2015; Kawamura and Bikkina, 2016; Nah et al., 2018). Although these atmospheric organic acids are typically weaker acids with higher $pK_a$ values than common inorganic acids ($H_2SO_4$ and $HNO_3$), they may contribute to particle acidity in some environments (Trebs et al., 2005). However, this effect is

not present in all environments and constrained to situations where the pH is in the range of the $pK_a$ of the acid in question (Nah et al., 2018; Song et al., 2018). As both organic acids and non-acid organic species are expected to be present, there are competing effects within the particle: dilution by the water fraction associated with organic constituents, direct acidification by the



dissolution of organic acids, and the change in $\gamma_{H+}$ by interactions with the additional species in solution.

Vasilakos et al. (2018) and Song et al. (2018) utilized E-AIM Model IV to evaluate the effect of oxalic acid on particle acidity. With inorganics similar in composition to that of Baltimore,

conditions applied in this study, the authors found that an increase of 25-50% of oxalic acid compared to the base case had an insignificant effect on aerosol pH when only one liquid phase was present. Pye et al. (2018) utilized data from the Southern Oxidant and Aerosol Study (SOAS) in ISORROPIA and AIOMFAC to investigate the gas-particle phase partitioning of ammonia, water, and organic compounds, and how liquid-liquid phase separation (LLPS) in

particles can affect aerosol pH, predicting a 0.7 pH unit increase when the organic fraction and its diluting effect was considered. Generally, aerosol processes are not affected by a pH change of this magnitude, except in the regions on the thermodynamic 'S-curves' of semi-volatile species where partitioning is shown to vary greatly for small changes in pH (Nah et al., 2018; Guo et al., 2018b). Additionally, owing to a combination of measurement uncertainties and

assumptions employed in the pH predictions, it is unlikely the aerosol pH can be accurately resolved to within less than 0.5 pH units of the real value.

In this work, we seek to explore the effects of WSOC on aerosol pH in a systematic way by utilizing inorganic data to construct combinations of hypothetical single, aqueous phase particulate compositions, and utilizing aerosol thermodynamic models to investigate the effects

of different WSOC species and concentrations on $\gamma_{H+}$ and $a_{H+}$.

## 2. Methods

**Data**



Inorganic and meteorological data used for this study were reported in prior work. Briefly, data
from Baltimore, MD were taken from Battaglia et al. (2017), and include speciated inorganic
$PM_{2.5}$ concentrations, meteorological data, and gas-phase $NH_3$ measurements. The data used as
thermodynamic model inputs are summertime (July) averages based on 3- or 5-years of

monitoring. All model inputs and outputs are available in the Supporting Information.

Aerosol inorganic composition, gas-phase $NH_3$ measurements, and meteorological parameters
were obtained during a study of winter haze formation in Beijing, China in 2015 (Wang et al.,
2016). These data represent a contrast with Baltimore due to different source contributions,
differences in $NH_3$ concentration (higher in Baltimore summer and Beijing winter), T, RH, and

inorganic concentration levels. The inorganic $PM_{2.5}$ concentrations, and averaged seasonal T and
RH, along with $NH_3$ gas concentration values are obtained as model-ready inputs of the Beijing
winter haze data from Guo et al. (2018b), based on supplemental information from Wang et al.
(2016).

**General Approach**

The general approach to this study was to utilize the inorganic PM and $NH_3$ data described
above, in combination with various additional WSOC constituents, as inputs to aerosol
thermodynamic equilibrium models to investigate the effects on model-predicted aerosol pH and
$\gamma_{H+}$. Inorganic data were modeled in either E-AIM IV or ISORROPIA-II to obtain equilibrium
concentrations of aerosol liquid water (ALW) along with all inorganic aerosol ionic species.

Organic constituents were then added to this invariant inorganic matrix (assuming the added
organic mass was at equilibrium), at identical T and RH, and the resulting particle compositions
were modeled in AIOMFAC to obtain aerosol $H^+$ ion activity ($a_{H+}$ and $\gamma_{H+}$), and thus aerosol pH.
The average inorganic composition, gas-phase $NH_3$, and meteorological conditions were held




approximately constant for each location, while WSOC composition and concentrations were

systematically varied. A matrix was constructed to examine multiple combinations of the

selected organic component composition levels (factorial design), and their effects evaluated on

the basis of organic-to-inorganic ratio (OIR) or organic mass fraction, both computed on a dry

5      particle basis. For each location, this resulted in a total of 7986 model simulations in total, with

1331 simulations run for both cases of organic compounds selected, and at each of three distinct

RH level, as described below. A summary of the models run for each location is shown in **Table
1**. The RH in all simulations was fixed at either ~70%, ~80%, or ~90%, with inorganic system

inputs calculated and invariant at each RH level based on the initial input data from either

Baltimore or Beijing to ensure deliquescence of inorganic aerosol particles, to understand the

sensitivity of the model-predicted aerosol pH to changes in RH (ALW), and to avoid liquid-

liquid phase separation as a potential cause of organic-influenced aerosol pH changes (Pye et al.,

2018).

**Organic Constituents**

Water-soluble organic compounds were selected by broadly classifying them as organic acids or

non-acid organics. Within each category, three individual species were selected based on

atmospheric relevance (*i.e.* are the species commonly encountered in whole or in part during

speciated organic studies?) and their presence in the standard table of AIOMFAC organic

species. In addition, non-acid organics were selected from three different primary moiety groups

from among the AIOMFAC standard species. Oxalic acid ($C_2H_2O_4$), glutaric acid ($C_5H_8O_4$), and

malonic acid ($C_3H_3O_4$) were selected as the three organic acid species. Levoglucosan ($C_6H_{10}O_5$),

tetrahydrofuran (($CH_2$)$_4$O), and 1-pentanol ($C_5H_{11}OH$) were selected as the non-acid organic

species. Concentration levels were not constrained by observations, but were instead selected to





achieve similar organic to inorganic mass ratios for each of the two geographic regions being considered. For each geographic region, 11 different concentrations were chosen (0-4 µg m$^{-3}$ for Baltimore; 0-40 µg m$^{-3}$ for Beijing), and combined in factorial fashion: each organic acid concentration level combination of the three organic acids, and vice versa for the non-acid

organic species. Combinations of organic acids and non-acid organic species were not explicitly considered here.

**Thermodynamic Model Input Configuration and Equilibrium Model Evaluations**

E-AIM Model IV provides thermodynamic equilibrium modeling of the $H^+$-$NH_4^+$-$Na^+$-$SO_4^{2-}$-$NO_3^-$-$Cl^-$-$H_2O$ system at temperatures from 263.15 K to 330 K for subsaturated systems that

contain $NH_4^+$ and $Cl^-$, or $Na^+$ in combination with other ions (Friese and Ebel, 2010). Data for Baltimore was formatted for E-AIM input in the following ways: average July inorganic species concentrations (µg m$^{-3}$) were converted to mol m$^{-3}$; the average daily temperature for the same period was used as the temperature input; the relative humidity of the system was fixed (at 70%, 80%, or 90%) both to ensure the inorganic system was in a deliquesced state and because of the

RH restrictions (subsaturated solution requirements, RH > 0.6) on E-AIM Model IV inputs. In addition to fixing system RH at 70%, 80%, or 90%, the aerosol metastable mode was enforced on the model by disabling the formation of all solids in the model input matrix, according to the analysis and recommendation of Guo et al. (2018b; 2015). Crustal species ($Ca^{2+}$, $Mg^{2+}$, $K^+$) not supported by the model were not considered, and the persistent cation deficiency was corrected

by the addition of $H^+$ to the system to ensure electroneutrality.

E-AIM offers support for certain organic acid species. For the Baltimore simulations, the organic acid species were added directly to the E-AIM model inputs. In the case of organic acid model runs, factorial combinations of the organic acid species at 0.0, 0.01. 0.02, 0.04, 0.08, 0.16,



0.32, 0.5, 1, 2, and 4 $\mu g\ m^{-3}$ were converted to mol $m^{-3}$ input. Formation of organic solids was also disabled as part of the equilibrium condition. For the non-acid organics, the addition of the selected species to the E-AIM equilibrium calculation was not possible, and the model was run with the inorganic constituents only.

## 5 ISORROPIA-II Model Input Configuration and Equilibrium Model Evaluations

ISORROPIA-II provides thermodynamic equilibrium modeling for the $H^+$-$NH_4^+$-$Na^+$-$SO_4^{2-}$-$NO_3^-$ -$Cl^-$-$Ca^{2+}$-$Mg^{2+}$-$K^+$- $H_2O$ across a wide range of temperature and RH values without limitation based on the input composition (Fountoukis and Nenes, 2007). Data for Beijing, China were already formatted for use in ISORROPIA-II as described above (Guo et al., 2018b). Metastable

mode was also used in this study, based on the justifications in previous studies (Guo et al., 2015; Guo et al., 2018b). An initial model run was performed to verify that identical model outputs were obtained using the inputs of Guo et al. (2018b). For the purposes of this investigation, the RH value was changed from the Beijing average ambient value of 56% to 70%, 80%, or 90%, consistent with the model input for the Baltimore data for the same reasons

discussed above. The Beijing average ambient temperature of 274.05 K was used in the Beijing inorganic model calculations with the three RH values.

### AIOMFAC Model Input Configuration and Equilibrium Model Evaluations

E-AIM and ISORROPIA-II were utilized to determine the equilibrium composition of the inorganic aerosol, including the $NH_3$ phase partitioning and the aerosol liquid water content.

Outputs from E-AIM and ISORROPIA were then used as inputs into AIOMFAC to characterize the organic effects on aerosol $H^+$ activity, and $\gamma_{H+}$.



The particle-phase outputs from the E-AIM and ISORROPIA-II model runs were used as inputs

to AIOMFAC; however, this required significant adjustments to the format to fit the AIOMFAC

model.  AIOMFAC requires inorganic species inputs to be entered as ionic *pairs* (whole

molecular species entered as a cation and anion pair) in order to guarantee electroneutrality.

Therefore, the ionic species outputs of E-AIM and ISORROPIA were converted to molecular

species inputs by assigning pairs, and then performing stoichiometric arithmetic until all ions

were accounted for (*i.e.* E-AIM $H^+$ and $SO_4^{2-}$ being combined in stoichiometric fashion as $H_2SO_4$

with corresponding reductions in the 'pool' of E-AIM $H^+$ and $SO_4^{2-}$).  In the Baltimore case for

the pure inorganic input (all organic species modeled at 0.0 $\mu$g m$^{-3}$ concentration), E-AIM Model

IV provided particle-phase output for the following ions: $H^+$, $NH_4^+$, $Na^+$, $HSO_4^-$, $SO_4^{2-}$, $NO_3^-$, and

$OH^-$.  In order to format these concentrations for AIOMFAC-specific inputs (that is, to compute

the necessary mole fraction format of molecular species in the aerosol), the ions were assigned in

the following ways.  First, all $SO_4^{2-}$ was associated with $H^+$ for the $H_2SO_4$ pair in AIOMFAC.

All $NO_3^-$ was associated with $Na^+$ for the $NaNO_3$ pair.  Remaining $Na^+$ was associated with $SO_4^{2-}$

, then $NH_4^+$ with $HSO_4^-$ and remaining $NH_4^+$ with the remaining $SO_4^-$.  This allocation process

proceeded similarly for the Beijing data.  The selected species and order of allocation of the ionic

species appears to be dependent solely on the researcher, and *a priori* knowledge of which

molecular species are likely to exist in the aerosol particle as the dissociated ionic species.  The

end result is a mixture of inorganic molecular species containing the full concentration values

generated by E-AIM or ISORROPIA assumed to be dissociated within the aerosol where each

functional group can contribute to species activity based on the AIOMFAC model paradigm;

assignment of molecular species pairings is performed only on the basis of formatting




specifically for the AIOMFAC model.   The inorganic inputs used in the AIOMFAC models for

both Baltimore and Beijing simulations are given in **Table 2**.

An additional key step in formatting the E-AIM and ISORROPIA output for input to AIOMFAC

is in the model treatment of the water associated with organic constituents, $W_o$.  E-AIM offers

limited support for organic compounds, and ISORROPIA offers no explicit support (*i.e.*, organic

species are not inputs to the model), so only in the case of Baltimore with organic acids did the

thermodynamic equilibrium models provide output of $W_o$ as a part of the total aerosol liquid

water (ALW; $W_i + W_o$).  RH is not an input to the AIOMFAC model runs.  Rather, AIOMFAC

requires the input of all species (inorganic and organic) in mole fractions, and assumes the

difference between the total inputs and unity is contributed by water, the water activity of which

is equal to the ambient fraction relative humidity.  Therefore, accounting for the water

contributed by the organic species was an additional step in formatting the E-AIM and

ISORROPIA outputs for AIOMFAC input as described below.  Water uptake predicted by the

three models was not directly compared for this reason.

For three of the four cases (Baltimore inorganics plus non-acid organics, Beijing inorganics plus

organic acids, and Beijing inorganics plus non-acid organics), $W_o$ was added to the system by the

following process, a flow diagram of which is shown in **Figure 1**.  For the first 11 points of the

factorial design (representing the addition of only the first organic constituent at each

concentration level) and the final 11 points of the factorial design (representing the 11 highest

organic addition points, including the addition of all three organic species at their maximum

selected concentration), total system moles were varied by increasing the inorganic model-

predicted moles of aerosol water.  AIOMFAC inputs (as mole fractions) were calculated using

this adjusted total mole value.  The 22 manually-adjusted points were modeled in AIOMFAC.  If





the option for liquid water is selected (as it was in all of our simulations), AIOMFAC assumes

that water makes up the difference between the mole or mass fraction of all inputs summed

together and unity. To achieve consistency with the inorganic model results, the total moles of

the system were required to be adjusted until the RH output generated by the AIOMFAC model

was within ~5% of the RH value fixed for the inorganic systems. Once this close fit was

achieved for the 22 selected points, they were used to generate polynomial fits of the total moles

added to the system as $W_o$ versus total organic mass (regardless of species). These polynomial

fits were then applied to all model points to adjust the total system moles through the addition of

liquid water associated with organic mass, resulting in AIOFAC-predicted RH values within 5%

of the E-AIM and ISORROPIA RH values of 70%, 80%, or 90%. This method of accounting for

$W_o$ is a strictly mathematical construct, and does not reflect the use of a species-dependent

organic hygroscopicity parameter, which would have been prohibitive to apply for each point

across all cases and RH levels. Additionally, following the introduction of the adjusted $W_o$ to the

system, the gas-phase was not allowed to re-equilibrate to the new water content contributed by

the organic species. This allows the effect of $W_o$ to have the highest possible magnitude (Guo et

al., 2015). All model inputs and outputs are included in the supplemental information.

**Thermodynamic Model Outputs**

Each model in this study was used to obtain specific relevant outputs. For E-AIM and

ISORROPIA, the models were used to obtain the aqueous-phase concentrations of the inorganic

species ($H^+$, $NH_4^+$, $Na^+$, $HSO_4^-$, $SO_4^{2-}$, $NO_3^-$, and $OH^-$), as well as $NH_3$ gas-phase concentration at

thermodynamic equilibrium (Figure 1). E-AIM was used for the Baltimore data to take

advantage of the support for organic acids, while ISORROPIA was used for the Beijing data to

allow for the use of (Guo et al., 2018b) as a verification on model output accuracy. AIOMFAC





was utilized to obtain the $H^+$ ion activity coefficient, and $H^+$ ion activity in the particle on a molality basis. The latter quantities are the outputs used to evaluate the influence of various organic compounds on the modeled-predicted pH and $\gamma_{H+}$. The AIOMFAC predicted system RH was used as a validity check for the equilibrium condition and input formatting.

## 3. Results and Discussion

**Effects of WSOC on Aerosol $\gamma_{H+}$**

AIOMFAC-predicted $\gamma_{H+}$ versus the organic dry mass fraction (total mass of organics/mass inorganics, excluding ALW), along with aerosol liquid water used in the model evaluations, are shown in **Figure 2** and **Figure 3** as well as **Figures S1-S4** in the Supplemental Information. For the case of organic acids at 80% RH (Fig. 2), ALW increases from $4.7 \times 10^{-9}$ L m$^{-3}$ to $1.4 \times 10^{-8}$ L m$^{-3}$ and from $9.2 \times 10^{-8}$ L m$^{-3}$ to $1.7 \times 10^{-7}$ L m$^{-3}$ for Baltimore and Beijing, respectively, as the organic mass fraction increases. This behavior makes sense, because the inorganic species concentrations and RH were fixed, so adding increasing levels of these organics increases the ALW. Increasing the organic dry mass fraction increased the value of $\gamma_{H+}$, from initial values of 0.10 and 0.16 for Baltimore and Beijing under inorganic-only conditions, to 0.65 for Baltimore (at an organic dry mass fraction of 0.77) and 0.62 for Beijing (organic dry mass fraction of 0.62). Note the higher absolute ALW levels in the Beijing simulations are due to the significantly higher inorganic and organic aerosol loadings.

The results follow for the additional RH values studied. For the case of organic acids at 70% RH (Fig. S1) increasing the organic dry mass fraction increases the value of $\gamma_{H+}$, from initial values of 0.12 and 0.18 for Baltimore and Beijing under inorganic-only conditions, to 0.84 for Baltimore (at an organic dry mass fraction of 0.76) and 0.79 for Beijing (organic dry mass





fraction of 0.63). ALW also increases when including the hygroscopic organic acid species,

going from $3.1 \times 10^{-9}$ L m$^{-3}$ to $8.6 \times 10^{-9}$ L m$^{-3}$ and from $5.9 \times 10^{-8}$ L m$^{-3}$ to $1.1 \times 10^{-7}$ L m$^{-3}$ for

Baltimore and Beijing, respectively. For the case of organic acids at 90% RH (Fig. S3) the effect

of increasing organic dry mass fraction is to increase the value of $\gamma_{H+}$, from initial values of 0.12

and 0.20 for Baltimore and Beijing under inorganic-only conditions, to 0.48 for Baltimore (at an

organic dry mass fraction of 0.77) and 0.49 for Beijing (organic dry mass fraction of 0.63). For

these simulations, the ALW increases from $9.0 \times 10^{-9}$ L m$^{-3}$ to $3.2 \times 10^{-8}$ L m$^{-3}$ and from $2.0 \times 10^{-7}$ L m$^{-3}$ to $3.8 \times 10^{-7}$ L m$^{-3}$ for Baltimore and Beijing, respectively.

For the case of the non-acid organics at 80% RH (Fig. 3), increasing the organic dry mass

fraction also increases the value of $\gamma_{H+}$ from initial values (as listed above for the inorganic-only

case) to 4.1 for Baltimore and 1.9 for Beijing. Aerosol liquid water also increases as expected,

from $4.9 \times 10^{-9}$ L m$^{-3}$ to $8.8 \times 10^{-9}$ L m$^{-3}$ and from $9.4 \times 10^{-8}$ L m$^{-3}$ to $1.5 \times 10^{-7}$ L m$^{-3}$ for

Baltimore and Beijing, respectively. For the case of non-acid organics at 70% RH (Fig. S2), the

behaviors were quite similar to the simulations at 80% RH, but the effects were amplified due to

the more concentrated particles. The $\gamma_{H+}$ values increased from 0.11 to 7.6 for Baltimore and

from 0.19 to 3.7 for Beijing. Aerosol liquid water also increases as expected. For the case of the

non-acid organics at 90% RH (Fig. S4) the behaviors of $\gamma_{H+}$ and ALW follow that at the lower

RH levels, but with higher ALW and lower maximum $\gamma_{H+}$.

The plots of aerosol liquid water display two distinct behaviors attributable to the way in which

the water content was derived for the model systems. For Baltimore with organic acid additions

(Figs. 2a, S1a, and S3a), the ALW was taken directly from the E-AIM Model IV output of

aqueous phase water (mol m$^{-3}$) since the organic acids were supported as built-in components for

the model. For the three remaining cases (Baltimore non-acid organics, both Beijing cases; Figs.



2b, S1b, and S3b; Figs. 3, S2, and S4), total ALW ($W_i + W_o$) was determined according to the

manual AIOMFAC output fitting/polynomial fit correlation described in the methods section

(Figure 1).

**Effects of WSOC on Aerosol pH**

Figures 2 and 3, along with Figures S1-S4, suggest differing effects of WSOC on aerosol pH. As

the dry organic mass fraction increased, ALW increased as well, since the RH and inorganics

were held constant. This suggests a diluting effect, which would increase pH, in agreement with

Guo et al. (2015). On the other hand, $\gamma_{H+}$ also increased with increasing dry organic mass

fraction, indicating that the addition of WSOC compounds increased the acidity (decreased pH).

AIOMFAC-predicted aerosol pH (molality basis) versus the dry inorganic mass fraction are

shown in **Figure 4** and **Figure 5**, as well as **Figures S5-S8** in the Supplemental Information for

each of the conditions considered. Aerosol pH was computed as the negative base-ten logarithm

of the hydrogen ion activity taken from the AIOMFAC output ($pH = -\log_{10} a_{H+}$) on a molality

basis. For the case of organic acids (Figs. 4, S5, and S7), increasing the organic mass fraction

results in only slight changes in the predicted aerosol pH. At 80% RH, the predicted pH ranges

from an initial (inorganic-only) value of 1.63 (Baltimore) and 5.5 (Beijing) to 1.61 (organic dry

mass fraction of 0.77) and 5.0 (organic dry mass fraction of 0.62). Similarly, there is a change

from 1.48 to 1.38 for Baltimore and a change from 5.12 to 5.04 for the Beijing data at 70% RH.

Finally, there is a change from 1.85 to 1.98 for Baltimore and a change from 5.1 to 5.14 for

Beijing at 90% RH.

The isolated group of points ($n = 53$) at pH = 5.0 in the 80% RH Beijing simulation arises from a

sudden discontinuity in the AIOMFAC-modeled $a_{H+}$ and $\gamma_{H+}$ between the organic dry mass

fractions of 0.464 and 0.467. This is not solely attributable to a discrete increase in the particle



concentration of any of the three organic acids (that is, it does not occur because oxalic, glutaric,

or malonic acids are increased from one concentration level to another, and thus starting a new

level in the factorial design scheme), but rather it occurs as oxalic acid transitions from 1 to 2 µg

m$^{-3}$ while glutaric acid and malonic acid remain constant at 2 and 4 µg m$^{-3}$m$^{-3}$, respectively. This

change is characterized primarily by an almost three-fold increase in the AIOMFAC-predicted

$a_{H+}$, from $3.6 \times 10^{-6}$ to $9.6 \times 10^{-6}$ and a similar increase in H$^+$ ion mass fraction ($6.2 \times 10^{-9}$ to $1.7$

$\times 10^{-8}$), mole fraction ($1.7 \times 10^{-7}$ to $4.5 \times 10^{-7}$), and molality ($8.4 \times 10^{-6}$ to $2.2 \times 10^{-5}$), while $\gamma_{H+}$

(0.431 to 0.433) and RH (80.7 % to 80.78%) remain mostly invariant at the transition between

these model points. Taken together, this may indicate that the system pH changes to such a point

that one of the acids which may not previously have been fully dissociated begins to do so,

spontaneously lowering system the pH by nearly 0.4 pH units.

For the case of non-acid WSOC additions (Figs. 5, S6, and S8), increasing the organic mass

fraction decreases the predicted aerosol pH from the initial inorganic-only values (listed above)

to a minimum of 1.2 (Baltimore) and 5.1 (Beijing) at 80% RH. For the 70% RH simulations, the

model predicted pH changes from 1.5 to 1.3 for Baltimore, and from 5.1 to 4.8 for Beijing. For

the 90% RH case, the model predicted pH changes from 1.8 to 1.6 for Baltimore, and from 5.1 to

5.0 for Beijing. The transition in the pH plots are smooth, where the contour lines reflect

individual levels of the factorial design and highlight the overall trend: as non-acidic WSOC is

added, AIOMFAC-predicted aerosol pH decreases for both the Baltimore and Beijing conditions.

Since the WSOC leads to ALW uptake (diluting acidity), the decrease in pH comes about due to

the increase in $\gamma_{H+}$.

**Figure 6** shows the AIOMFAC-modeled aerosol pH for the pure WSOC species under the

Baltimore and Beijing conditions at 80% RH. These simulations are a subset of those shown in





Figure 2. Figure 6 shows that the effect of WSOC on $\gamma_{H_+}$ is greater than its effect on ALW, since pH systematically decreases with WSOC addition for all compounds in Beijing, and all compounds except levoglucosan in Baltimore. When the two effects are combined, as they are in the model cases where organic species are added together in factorial fashion, the overall

nearly-invariant pH behavior shown in the previous figures is observed. Additionally, it can be seen that even for the case of pure-species WSOC addition, the observed range of the pH change is limited to < 0.5 pH units. The magnitude of this pH change is not expected to significantly alter particle conditions or lead to changes in particle chemistry, except when the pH is close to the point where a given species is almost equally partitioned between the gas and particle phases.

This effect can been clearly in the work of Guo et al. (2018b) and Vasilakos et al. (2018), who demonstrate this effect by the use of 'S-curves' for species partitioning. When the pH lays on or near the inflection point of the S-curve, a change of 0.5 pH units will have significant effect on species partitioning; however, when the pH is in the near-horizontal portions of the curve above or below the rapid transitional region, a change of 0.5 pH units will have negligible effect on

partitioning, and thus particle chemistry.

Taken together, these results indicate that, despite organic mass fractions greater than 60% (dry particle mass basis), the combined effects of WSOC species on model-predicted aerosol pH is only about 0.5 pH units, maximum, with most pH changes < 0.2 pH units. The effect is even more pronounced in the case of non-acid organics, which is a surprising and somewhat

counterintuitive result. Addition of soluble species increases $\gamma_{H_+}$ in solution (and thereby $a_{H_+}$, lowering pH), while the hygroscopic nature of the species adds additional ALW with a diluting effect (increasing pH). This result is observed for organic acids and non-acid species alike, and for simulations with a single organic compound added or for mixtures (**Table 2**). This suggests



that the overall effect of WSOC on aerosol pH is quite minimal in conditions where LLPS does

not occur. This finding holds only for systems in which there is no LLPS and the solvent is $H_2O$,

where the definition of pH holds. For systems in which LLPS does occur, a condition expected

in systems with O:C ratio of the organic material $\leq 0.5$, or RH < 60% with organic:sulfate mass

ratio < 1 (Bertram et al., 2011; You et al., 2013), the situation becomes more complicated as

there exists no accepted definition of pH for a predominantly-organic phase in which the solvent

is the organic aerosol constituents rather than ALW. However, this is not anticipated in either

Baltimore or Beijing, where fully-SOA particles are anticipated to remain liquid to atmospheric

heights of ~2km in the planetary boundary layer (Shiraiwa et al., 2017).

This work stands apart from, but connects to related works. Pye et al. (2018) specifically

examined the effects of LLPS, but the present study examines a different particle regime

altogether (single aqueous phase with water, inorganics, and organics). Guo et al. (2018a)

demonstrated that organics do not affect ammonia partitioning (and thus acidity) in cases

consistent with LLPS. We believe that these results are part of related physical phenomena. In

the LLPS case, organic components and their associated water form a phase that is distinct from

the predominantly aqueous-inorganic phase. This results in less water in the aqueous phase

being available to counteract the effect of inorganic species equilibrium processes ($SO_4^{2-}$–$HSO_4^-$

equilibrium; $NH_3$-$NH_4^+$ equilibrium), with a resulting effect on modeled aerosol pH. LLPS, and

the associated effects on modeled aerosol pH may dominate at low RH; however, as RH

increases, the impact of LLPS is lessened. Where the two phases finally become miscible, $W_o$

and the organic components previously in a distinct organic phase become accessible to the

inorganic mixtures in the aqueous phase, resulting in the minimal effects on modeled aerosol pH

demonstrated here as a result of competing dilution and $\gamma_{H+}$ effects.



Our findings are also supported by the work of Song et al. (2018), who utilized E-AIM Model IV and ISORROPIA to  model the same Beijing winter haze conditions, and found that addition of oxalate (as oxalic acid) to their model in E-AIM produced reductions in pH of only 0.07 pH units.  Our results are also consistent with those of Vasilakos et al. (2018), who observed a

similarly minor effect of oxalate addition on aerosol pH in the Eastern U.S., and Nah et al. (2018) where oxalic acid/oxalate gas-particle partitioning predicted without considering organic species in the thermodynamic analysis was in reasonable agreement with measurements.

A limitation of this study is that the model simulations were only run at three RH levels (70%, 80%, and 90%), with metastable conditions enforced at all times.  However, aerosol particles

progress through a greater RH range in the atmosphere, with concomitant effects on aerosol liquid water and phase transitions.  Future works would need to expand on the RH range in order to elucidate the behavior as the system transitions from the LLPS condition to the fully mixed aqueous condition, and the contribution of changing ALW.  Additionally, the use of E-AIM Model IV imposes composition limitations on the inputs (i.e., no support for $Ca^{2+}$, $Mg^{2+}$, or $K^+$;

limited support for $Na^+$ in the presence of $NH_4^+$ and $Cl^-$), necessitating the use of equivalent cations to maintain electroneutrality in the model inputs.  Combined with the use of metastable calculations, there exists a potential source of error in the solution activity if these species are considered and allowed to precipitate out in the thermodynamic model calculations (e.g., $CaSO_4$).  As AIOMFAC relies on specific, uniquely-defined functional group interactions in the

composition of activity coefficients, the exchange of a non-supported cation in E-AIM for a charge-equivalent cation may have effects on the output unknown to us carried through to the calculation of the species activity coefficients in AIOMFAC; this is a limitation of



thermodynamic models that has been previously discussed (Jacobson, 1999; Kim and Seinfeld, 1995).

A significant limitation of our study is consideration of only six WSOC species, despite hundreds or thousands being present in atmospheric particles. This is a limitation we

acknowledge, but is based on the significant number of model runs given the factorial design paradigm, and the decision to utilize only compounds predefined in the thermodynamic models (particularly the AIOMFAC model, which allows users to create organic molecules by combining subgroups). Because the compounds selected here have relatively low molecular weight (MW), is it possible that higher MW compounds, such as humic-like substances

(HULIS), may impart a different effect. However, given the consistent results found here for both Baltimore and Beijing conditions, across the 70-90% RH range, and at organic dry mass fractions that range from 0 - 60% utilizing WSOC containing four moieties, we feel our results do represent conditions in atmospheric particles. Future studies would be necessary to expand the selection of WSOC compounds, and thus broaden the results reported here. Because we have

forced the metastable condition on our use of the models in order to avoid any potential LLPS, the system mixing state becomes another potentially significant source of error. Here we have considered only internally mixed aerosol particles without LLPS, a case that may not exist given the concentration of organic species utilized in the model study. The most significant restriction of this study is the lack of observational data for comparison. Direct measurements of particle

pH have so far been restricted to simple laboratory particles of specific super-micron sizes and compositions (Rindelaub et al., 2016).

## 4. Conclusions and Implications





In this work, the effects of WSOC on model-predicted aerosol pH were evaluated. Different

inorganic datasets from Baltimore and Beijing winter haze conditions representing distinct

inorganic composition regimes were first modeled in aerosol thermodynamic equilibrium models

(E-AIM or ISORROPIA), then combined with six different organic species in AIOMFAC to

determine the effects on aqueous-phase $\gamma_{H+}$ and $a_{H+}$. We find that the effects of both organic

acids and non-acid WSOC species addition to each of the regions has only a modest effect on

aerosol pH ($< 0.5$ pH units, with most $< 0.2$ pH units). These small effects on pH were predicted

even up to organic dry mass in excess of 60%. The magnitude of these changes to aerosol pH

are consistent with the results predicted by previous studies that considered only inorganic

aerosol components combined with *a priori* knowledge of organic mass, organic water

contribution, and organic species hygroscopicity (Guo et al., 2015; Bougiatioti et al., 2016).

The results of this study have important implications, for the aerosol modeling community as

well as for experimental studies that utilize phase partitioning data to constrain aerosol pH.

Previous studies have postulated on the effect of organic species while ignoring their inclusion,

or included them in order to elucidate the effects of LLPS, but this study demonstrates that in the

case of single-phase systems, including these species may only contribute unnecessary

complexity to the model runs. As their effects are demonstrated to be $< 0.5$ pH units (and most

frequently $< 0.2$ pH units), it is not expected that the inclusion of organics will cause the pH of

the system to reach any significant transitions unless the organic components have already driven

the system to a sensitive portion of the species partitioning S-curves, for especially glassy

aerosols, or aerosols in which there is significant phase separation. This work demonstrates that

inclusion of large quantities of organic components does not appear to have a significant effect

on model-predicted aerosol pH, consistent with the findings of Guo et al. (2018a) and Vasilakos





et al. (2018). Based on the species and concentrations of the organics studied here, future
aerosol modeling studies carried out in areas with significant amounts of lower molecular weight
organic species may be justified in the use of inorganic-only aerosol thermodynamic equilibrium
models to predict aerosol pH without the direct inclusion of organic species.

## 5. Author Contribution

CH and MB conceived of the study. MB performed all of the modeling analyses. MB, RW, AN,
and CH collaborated on the data interpretation and all contributed to manuscript writing and
editing.

## 6. Acknowledgements

M.B and C.H. were supported by NSF Grant # CHE-1454763. A.N. acknowledges support from
the project PyroTRACH (ERC-2016-COG) funded from H2020-EU.1.1. - Excellent Science -
European Research Council (ERC), project ID 726165.

## 7. Data Availability

All model inputs and outputs have been published and are freely available through the Maryland
Shared Open Access Repository: http://hdl.handle.net/11603/13532





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

value (70%, 80%, or 90%). These 22 points were then used to fit a polynomial function to

correct the total system moles for the remaining 1309 data points.



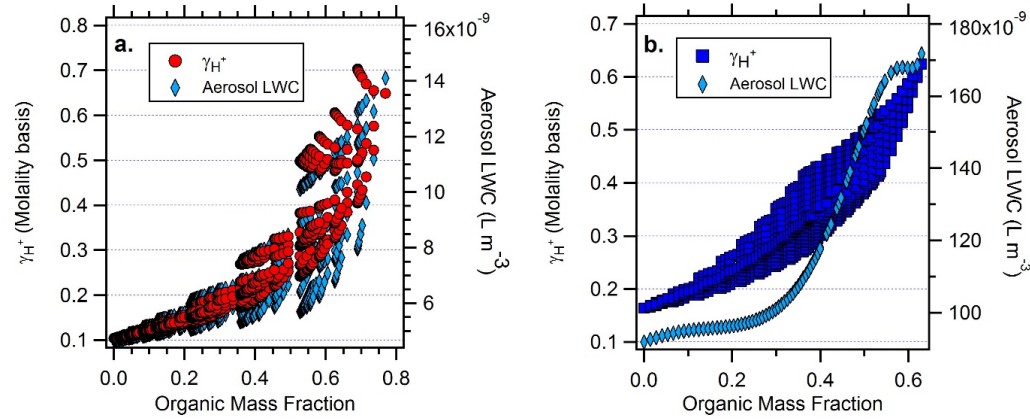

**Figure 2**:  AIOMFAC-modeled $\gamma_{H^+}$ (molality basis) and aerosol liquid water (ALW, E-AIM

Model 4 output or polynomial fit to AIOMFAC output) versus organic dry mass fraction with the

factorial addition of organic acid species for a) Baltimore and b) Beijing at 80% RH.



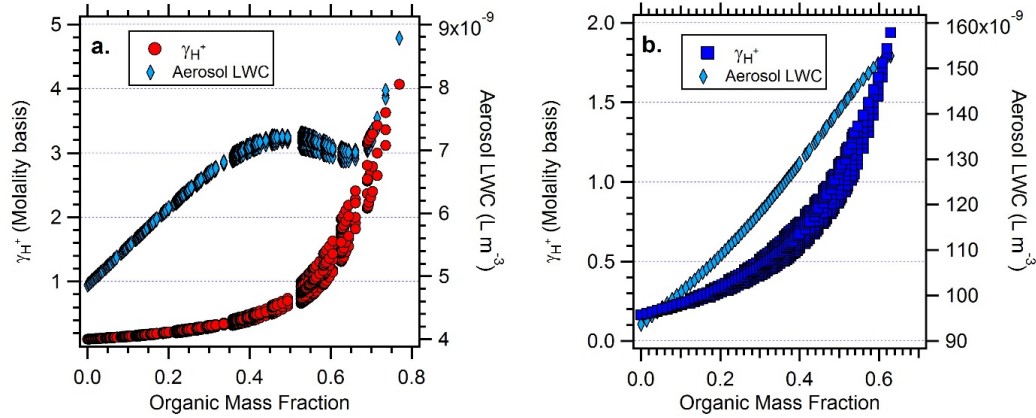

**Figure 3**: AIOMFAC-modeled $\gamma_{H+}$ (molality basis) and aerosol liquid water (ALW, polynomial

fit to AIOMFAC output) versus organic dry mass fraction with the factorial addition of non-acid

organic species for a) Baltimore and b) Beijing at 80% RH.





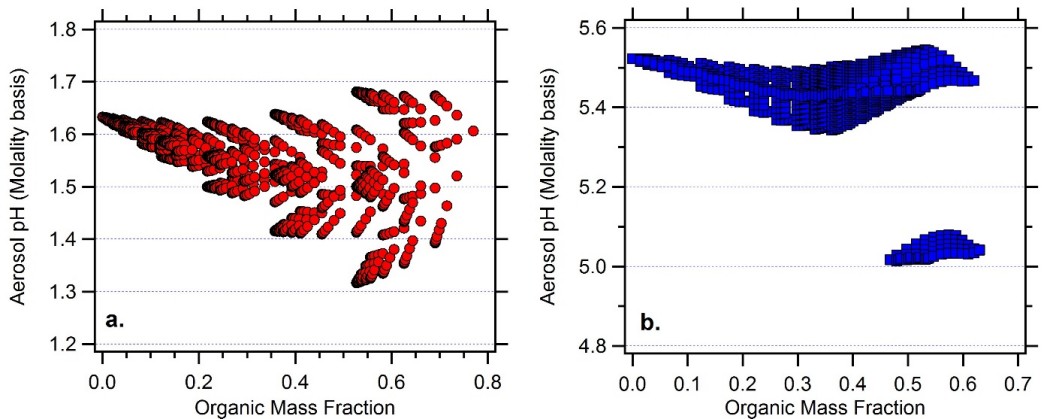

**Figure 4**:  AIOMFAC-modeled aerosol pH (molality basis) versus organic dry mass fraction

with the factorial addition of organic acid species for a) Baltimore and b) Beijing at 80% RH.





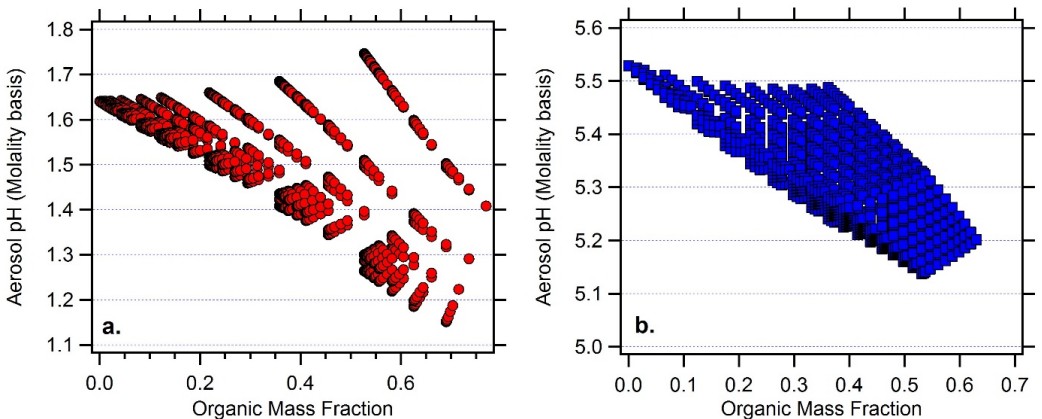

**Figure 5**: AIOMFAC-modeled aerosol pH (molality basis) versus organic dry mass fraction with

the factorial addition of non-acid organic species for a) Baltimore and b) Beijing at 80% RH.

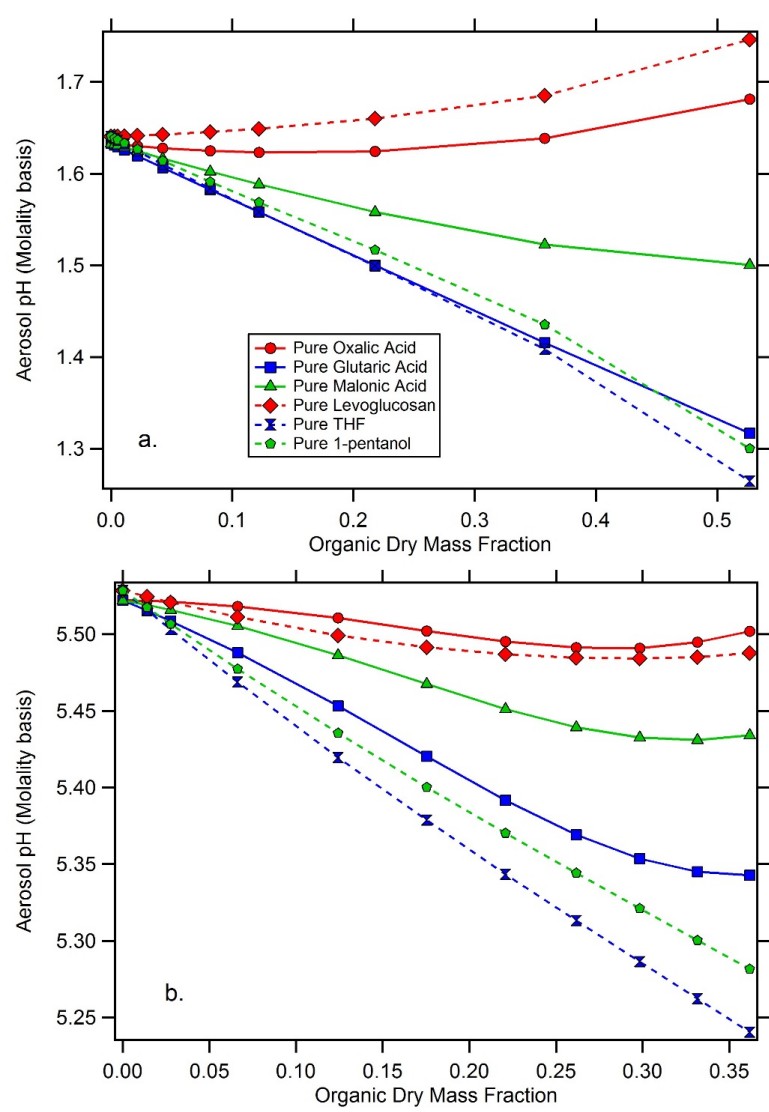

**Figure 6**: AIOMFAC-modeled aerosol pH (molality basis) versus organic dry mass fraction

displayed for each of the six pure species simulation runs for a) Baltimore and b) Beijing.



**Table 1**: Summary of model runs performed for the study. Inorganic composition is invariant

and taken from the sources provided. Organic components were added in factorial fashion.

| Location | Inorganic Data | Inorganic Equilibrium Model | Organic components | # of Points |
|---|---|---|---|---|
| Baltimore | Battaglia et al. 2017 | E-AIM Model IV | Organic Acids | 1331 |
| Baltimore | Battaglia et al. 2017 | E-AIM Model IV | Non-acid organics | 1331 |
| Beijing | Guo et al. 2018b | ISORROPIA 2.3 | Organic acids | 1331 |
| Beijing | Guo et al. 2018b | ISORROPIA 2.3 | Non-acid organics | 1331 |



**Table 2**: Whole species inorganic inputs used in AIOMFAC modeling at 70%, 80%, or 90% RH.  Inorganic equilibrium outputs were

used to assign anions to cations to form whole species, represented here.  Entries marked with a dash represent components not used

for the given location as a result of anion or cation not being present in the composition data (species not measured).

| Component | Baltimore Conc. (mol m⁻³) 70% RH | Baltimore Conc. (mol m⁻³) 80% RH | Baltimore Conc. (mol m⁻³) 90% RH | Beijing Conc. (mol m⁻³) 70% RH | Beijing Conc. (mol m⁻³) 80% RH | Beijing Conc. (mol m⁻³) 90% RH |
|---|---|---|---|---|---|---|
| $H_2SO_4$ | $1.488 \times 10^{-10}$ | $2.411 \times 10^{-10}$ | $3.326 \times 10^{-10}$ | $1.718 \times 10^{-12}$ | $1.127 \times 10^{-12}$ | $3.106 \times 10^{-12}$ |
| $NH_4HSO_4$ | $3.460 \times 10^{-9}$ | $2.612 \times 10^{-9}$ | $1.899 \times 10^{-9}$ | $8.925 \times 10^{-13}$ | $2.470 \times 10^{-13}$ | $5.139 \times 10^{-12}$ |
| $(NH_4)_2SO_4$ | $2.395 \times 10^{-8}$ | $2.318 \times 10^{-8}$ | $2.425 \times 10^{-8}$ | $2.651 \times 10^{-7}$ | $2.653 \times 10^{-7}$ | $2.651 \times 10^{-7}$ |
| $NaNO_3$ | $6.425 \times 10^{-11}$ | $1.207 \times 10^{-10}$ | $3.573 \times 10^{-10}$ | - | - | - |
| $Na_2SO_4$ | $1.403 \times 10^{-9}$ | $1.375 \times 10^{-9}$ | $1.257 \times 10^{-9}$ | - | - | - |
| $NaCl$ | - | - | - | $1.000 \times 10^{-20}$ | $1.000 \times 10^{-20}$ | $1.000 \times 10^{-20}$ |
| $NH_4Cl$ | - | - | - | $4.659 \times 10^{-8}$ | $4.646 \times 10^{-8}$ | $4.663 \times 10^{-8}$ |
| $NH_4NO_3$ | - | - | - | $4.125 \times 10^{-7}$ | $4.126 \times 10^{-7}$ | $4.126 \times 10^{-7}$ |

54