# Peer review of "Effects of Water-soluble Organic Carbon on Aerosol pH"

_Atmospheric Chemistry and Physics, 2019_

## Referee Comment (RC1) · Anonymous Referee #1 · 10 Jun 2019

This article by Battaglia, Weber, Nenes and Hennigan explores effects of water-soluble organic compounds on aerosol acidity by means of thermodynamic modeling. Two data sets from field measurements distinct by acidity (Baltimore & Beijing) are used to provide information about inorganic constituents, aerosol mass concentration and the water-soluble organic fraction. The authors provide new insights into the effects of organic compounds on aerosol water content and the activity coefficient of H+. This topic is of interest for the atmospheric chemistry community, since aerosol acidity has been linked to a range of chemical and physical properties of particulate matter, but relatively few studies have attempted to quantify the effects of organic/inorganic mixing on aerosol pH. While certain effects of inorganic/organic aerosol mixing have been excluded in this work, e.g. liquid-liquid phase separation and lower relative humidity

conditions, the study by Battaglia et al. provides interesting findings for single-phase aqueous aerosol by using a combination of state-of-the-art thermodynamic models. It is shown for the cases studied that the pH is largely controlled by the inorganic aerosol species, which is in agreement with other related studies (Pye et al., 2018).

The manuscript is generally well structured and provides a good introduction about the need for understanding the effects of organic acids and non-acids on pH. However, the descriptions lack some details with regard to the methods employed and would benefit from additional discussion about the choice of a rather limited number of specific organic non-acids and dicarboxylic acids and how these may affect the simulation results.

I am in support of the publication of this work after my minor points on methods and assumptions have been addressed by the authors. General and specific comments are provided below.

**1 General comments**

1. Factorial design method (pages 8, 9).
   A factorial modeling experiment design was used to represent different combinations and concentration levels of organic compounds. Since this is not a widespread approach used by atmospheric chemists, it would be adequate to provide a better description of what was done exactly and why. For example, on page 9 is stated: "... *and combined in factorial fashion: each organic acid concentration level combination of the three organic acids, and vice versa for the non-acid organic species. Combinations of organic acids and non-acid organic species were not explicitly considered here.*"
   This could be phrased much better for clarity and perhaps an example could be provided to make clear what is meant (e.g. that the approach involves indepen-

dent combinations, rather than permutations) and what is "factorial" about this method. Also, references to adequate literature describing this approach are missing.

2. Liquid-liquid phase separation (LLPS).
Phase separation was not considered in the calculations from this study, but the topic is discussed in the introduction, which is valuable. It is well-known that non-ideal mixing in organic/inorganic solutions can cause LLPS, as the authors also point out. However, what remains unclear is how the authors concluded that their cases are not affected by phase separation. Page 8, line 8 – 12: "*The RH in all simulations was fixed at either 70%, 80%, or 90%, with inorganic system inputs calculated and invariant at each RH level based on the initial input data from either Baltimore or Beijing to ensure deliquescence of inorganic aerosol particles, to understand the sensitivity of the model-predicted aerosol pH to changes in RH (ALW), and to avoid liquid-liquid phase separation as a potential cause of organic-influenced aerosol pH changes (Pye et al., 2018).*"
It should be clarified how avoiding LLPS is related to deliquescence of inorganic particles. How sure are the authors that LLPS would not occur, say at 70% RH, in their systems when high amounts of organics are present? Was this checked quantitatively? For example, the authors could confirm that organic activity coefficients remain less than $\sim 5 - 10$, since studies like Donahue et al. (2011, https://doi.org/10.5194/acp-11-3303-2011) suggest that "Somewhere between $5 < \gamma < 10$ phase separation becomes very likely". The LLPS constraints from You et al. 2013 (mentioned much later on page 19) could also be useful for the discussion in this context.

3. Acids vs. non-acid organics.
In the abstract and similarly on page 18, line 19, it is stated that "*surprisingly, non-acidic WSOC compounds were found to have a larger effect on pH than organic acids owing to their stronger impacts on $\gamma_{H+}$*". It is not made clear what

is *surprising*, please clarify. Also, how robust is this result given the choice of rather small, polar and hygroscopic organic diacids in this study. Would the same explanation hold true if a diacid of lower O:C ratio (less hygroscopic) were used? Moreover, the finding may depend on how ISORROPIA, E-AIM, UNIFAC and AIOMFAC treat organic acids (accounting for dissociation or not); see specific comments.

**2  Specific comments**

- Page 2, line 23: define the meaning of "*aerosol strong acidity or total acidity*"

- P3, l6: "*to obtain the pH of an aerosol distribution*". Since distribution is mentioned, would the same pH be expected for all aerosol particles over a considerable size range (e.g. ultrafine vs. larger accumulation mode particles)? Do the input data from measurements account for size modes?

- P3, l10 - 15: the aerosol thermodynamic models are introduced. At this point it is appropriate to define the acronyms and to cite the key references describing the models, which are missing; see associated websites. Also, citing Ganbavale et al. (2015) in the context of AIOMFAC and its web model seems inadequate as that study does not concern organic-inorganic mixtures.

- Pages 3 and 4 (first paragraph): AIOMFAC is discussed as only being an activity coefficient model, which seems correct, but the authors forgot to mention that there are equilibrium models based on AIOMFAC, including the one used by Pye et al. (2018), Hodas et al. (2015), Zuend et al. (2010, 2012) and others. The study by Pye et al. used both ISORROPIA and AIOMFAC-based equilibrium models to estimate aerosol pH. It seems that the authors only refer to the online model of AIOMFAC, which could be made clear. Also, do ISORROPIA, E-AIM,

AIOMFAC solve the ion association/dissociation equilibria, like bicarbonate and bisulfate in aqueous systems? What about organic acid dissociation? There is an E-AIM model version for specific acids, but it seems that organic acid dissociation is not considered explicitly by the models, right? Discussion of this could be important for pH calculations in the context of diacids (of different pKa values) and when contrasting the effects of acidic vs. non-acidic organics.

- P4, l 15: Define LLPS

- P5, l 5: Rephrase "increasing particle phase partitioning"; what kind of partitioning is meant? gas-particle, liquid-liquid?

- P6, l12: Rephrase "S-curves" - these are known as sigmoid curves.

- P6, l16: how was the threshold uncertainty of 0.5 pH units determined? Cite a reference.

- P8, l18: What is "the standard table of AIOMFAC organic species"? this is unclear

- P10, l9: Clarify meaning of "Metastable mode" in this context. This seems to be a technical term used with ISORROPIA, but jargon should be avoided.

- P11, l16: "*The selected species and order of allocation of the ionic species appears to be dependent solely on the researcher, and a priori knowledge of which molecular species are likely to exist in the aerosol particle as the dissociated ionic species.*" It is unclear what a priori knowledge is needed. Do the authors mean that the choice of species allocation affects model predictions or not?

- P17, l1 - 4: The statements in this sentence appear to contradict each other. Is the increase of oxalic acid from 1 to 2 $\mu$g/m$^3$ not a discrete increase in concentration? Revise.

- P17, l9 - 11: "*Taken together, this may indicate that the system pH changes to such a point that one of the acids which may not previously have been fully dissociated begins to do so, spontaneously lowering system the pH by nearly 0.4 pH units*". Are you referring to inorganic acids here or to the dicarboxylic acids? If the latter, is AIOMFAC even computing the dissociation of acids (explicitly or implicitly)?

- P18, l10: rephrase

- P19, l5: "*the situation becomes more complicated as there exists no accepted definition of pH for a predominantly-organic phase in which the solvent is the organic aerosol constituents rather than ALW.*"
  This referee considers this statement to be incorrect. Isn't the definition by IUPAC the accepted definition of pH, which was also used in this study? The definition of pH by IUPAC seems to be applicable in either situation; it solely depends on the H+ activity. Further, any (organic) phase containing dissolved H+ likely also contains some amount of water.

- P19, l10: Didn't Pye et al. (2018) consider also cases with a single aqueous phase? Their Figure 3 suggest they had both LLPS and single phase cases for the SOAS field study.

- P21, l15: It is unclear how a "forced metastable condition" would help to avoid LLPS. These are two independent processes/states. Clarify.

- P22, l20: "for especially glassy aerosols" – unclear how viscosity and glassy aerosols are related to pH in this context. Glassy aerosols would indicate significant equilibration time scale effects, but what does that have to do with pH?

- P23, l4: While the authors show that the aerosol pH value is mostly determined by inorganic species, the present study and previous work cover a relatively limited

set of organics compounds and conditions. Hence, general conclusions should be stated carefully. One aspect that warrants further study and discussion concerns the partitioning of organic acids that may dissociate significantly at relatively high pH (say > 4) as well as the effects of amines acting as bases similar to ammonia. Therefore, it may be premature to conclude that exclusion of organics for future aerosol pH predictions is always justified. A disclaimer in that context may be appropriate.

---

## Referee Comment (RC2) · Anonymous Referee #2 · 12 Jul 2019

In "Effects of Water-soluble Organic Carbon on Aerosol pH", the authors examine changes in pH due to water solute organic acid with inorganic-organic single-phase aerosol systems, at conditions from two locations: Baltimore (more acidic) and Beijing (less acidic). In the manuscript, the authors detail use of three widely used thermodynamic models: E-AIM, ISORROPIA, and AIOMFAC for calculations of water content, activity coefficients, and pH. The results should that the impact of the organic content explored here on pH was minimal, and continues to support the role of the inorganic compounds in determining pH. However, there are some unanswered questions regarding the calculation methods, and in particular the role of dissociation equilibria of the organic acids, that should be clarified or handled in more depth. The paper would be of interest to the Atmospheric Chemistry and Physics readership, and should be

considered for publications after the following are addressed.

1. Organic acids and dissociation:

It was stated that "non-acidic WSOC were found to have a larger effect... than organic acids" due to their larger impact on H+ activity. However, it is not clear how the models used handle the concentration and the concentration, ionic strength, and pH dependent dissociation of the weak organic acids. Clarify how AIOMFAC treats the organic acids. Selection of organic acids can be put into E-AIM, both with an UNIFAC treatment, and a fitted treatment, which, I believe, treats the organic acid as non-dissociating and partially-dissociating. It would be helpful to see the impact of the two treatments in the pH results. It may be that different treatment of the dissociation equilibrium may result in more significant changes in pH by the organic acid.

2. Effects due to selected method/models:

There remains an underlying question of if the results (e.g., the minimal impact of WSOC on aerosol pH) is due to the method and models used, versus being representative of the true pH in the Baltimore and Beijing conditions. It would be helpful to compare results with various treatments of organic acid dissociation equilibrium (see above comment) as well as organic-inorganic mixtures. For the later, one can add organic compounds to E-AIM, by way of using UNIFAC functional groups – how do the results from this method compare to the that with a combined E-AIM for the inorganic, AIOMFAC for the organic method used here? E-AIM was used for the Baltimore date, and ISORROPIA for the Beijing data – how do the results vary if this was switched? For example, a model-based impact likely explains the isolated group of points in Fig 4 (page 16 line 21 – page 17 line 11) – is that same group of points found with oxalic acid is treated using a different model?

3. Effects of the inorganic salts and the model organics chosen:

How much impact did the choice of method to go from individual ions to salts (page

10.5194/acp-2019-344

11) have on the results? How does this compare to results if the ions were evenly distributed into all the possible salt combinations? For the organics, more information on why the three particular non-acids were chosen. (line 22, page 8) Also, why were organic acids and non-organic acids not considered together? (line 5, page 9). Is it expected that having the mixture would impact the results?

4. Minor comments:

- Define what is meant by "... the thermodynamic equilibrium problem..." (line 2, page 4).

- What are the typical organic to inorganic mass ratios of the two region (line 1, page 9)

- How much H+ was typically added for electroneutrality (line 19-20, page 9), compared to the other cations? Was it significant? Please comment.

- "AIOFAC" typo, line 9, page 13
* * *

---

## Author Comment (AC1) · 22 Sep 2019

**Response to Referee Report**

We thank the Referees for their helpful suggestions and detailed comments. We have enumerated the points made by the referees below, and provided our responses to each point in blue text. We have also provided a revised manuscript with changes highlighted (starting on pg. 13 of this file).

**Referee #1**

This article by Battaglia, Weber, Nenes and Hennigan explores effects of water-soluble organic compounds on aerosol acidity by means of thermodynamic modeling. Two data sets from field measurements distinct by acidity (Baltimore & Beijing) are used to provide information about inorganic constituents, aerosol mass concentration and the water-soluble organic fraction. The authors provide new insights into the effects of organic compounds on aerosol water content and the activity coefficient of H+. This topic is of interest for the atmospheric chemistry community, since aerosol acidity has been linked to a range of chemical and physical properties of particulate matter, but relatively few studies have attempted to quantify the effects of organic/inorganic mixing on aerosol pH. While certain effects of inorganic/organic aerosol mixing have been excluded in this work, e.g. liquid-liquid phase separation and lower relative humidity conditions, the study by Battaglia et al. provides interesting findings for single-phase aqueous aerosol by using a combination of state-of-the-art thermodynamic models. It is shown for the cases studied that the pH is largely controlled by the inorganic aerosol species, which is in agreement with other related studies (Pye et al., 2018). The manuscript is generally well structured and provides a good introduction about the need for understanding the effects of organic acids and non-acids on pH. However, the descriptions lack some details with regard to the methods employed and would benefit from additional discussion about the choice of a rather limited number of specific organic non-acids and dicarboxylic acids and how these may affect the simulation results.
I am in support of the publication of this work after my minor points on methods and assumptions have been addressed by the authors. General and specific comments are provided below.

General Comments
1.) Factorial design method (pages 8, 9).
A factorial modeling experiment design was used to represent different combinations and concentration levels of organic compounds. Since this is not a widespread approach used by atmospheric chemists, it would be adequate to provide a better description of what was done exactly and why. For example, on page 9 is stated: "...and combined in factorial fashion: each organic acid concentration level combination of the three organic acids, and vice versa for the non-acid organic species. Combinations of organic acids and non-acid organic species were not explicitly considered here."

This could be phrased much better for clarity and perhaps an example could be provided to make clear what is meant (e.g. that the approach involves independent combinations, rather than permutations) and what is "factorial" about this method. Also, references to adequate literature describing this approach are missing.

We have added clarifying statements as to the nature and structure of the factorial design method, giving attention to the statistical design terms of 'factors' and 'values.' Additionally, we have provided a highly-cited textbook on statistical design that discussed factorial design of experiments. The updated text (lines 187-193) now reads, "A matrix was constructed to examine multiple combinations of the selected organic component composition levels (factorial design), and their effects evaluated on the basis of organic-to-inorganic ratio (OIR) or organic mass fraction, both computed on a dry particle basis. This full factorial design consists of three factors for each acid or non-acid condition (the identity of each species), each with discrete possible values (air concentrations in µg m$^{-3}$), where the experiment incorporates all possible combinations of these values across all factors (Keppel, 1991)."

And (lines 220-226): "For each geographic region, 11 different concentrations were chosen for each WSOC compound (0-4 µg m$^{-3}$ for Baltimore; 0-40 µg m$^{-3}$ for Beijing), and combined in factorial fashion: each organic acid concentration level combination of the three organic acids were examined in combination with every other level of the remaining two, and vice-versa for the non-acid organic species. Combinations of organic acids and non-acid organic species were not explicitly considered here; only combinations of organic acids with organic acids, or combinations of non-acids with non-acids were examined experimentally."

2.) Liquid-liquid phase separation (LLPS)
Phase separation was not considered in the calculations from this study, but the topic is discussed in the introduction, which is valuable. It is well-known that non-ideal mixing in organic/inorganic solutions can cause LLPS, as the authors also point out. However, what remains unclear is how the authors concluded that their cases are not affected by phase separation. Page 8, line 8 – 12: "The RH in all simulations was fixed at either 70%, 80%, or 90%, with inorganic system inputs calculated and invariant at each RH level based on the initial input data from either Baltimore or Beijing to ensure deliquescence of inorganic aerosol particles, to understand the sensitivity of the model-predicted aerosol pH to changes in RH (ALW), and to avoid liquid-liquid phase separation as a potential cause of organic-influenced aerosol pH changes (Pye et al., 2018)."

It should be clarified how avoiding LLPS is related to deliquescence of inorganic particles. How sure are the authors that LLPS would not occur, say at 70% RH, in their systems when high amounts of organics are present? Was this checked quantitatively? For example, the authors could confirm that organic activity coefficients remain less than ~5 – 10, since studies like

Donahue et al. (2011, https://doi.org/10.5194/acp-11-3303-2011) suggest that "Somewhere between 5< $\gamma$ <10 phase separation becomes very likely". The LLPS constraints from You et al. 2013 (mentioned much later on page 19) could also be useful for the discussion in this context.

The referee brings up an excellent point. We have addressed the LLPS condition for the non-acid simulations quantitatively in revision in the following ways: using the parameterization of Bertram et al. ACP 2011, who define separation RH (SRH) as a function of O:C, we compute the mass-weighted O:C for all of our simulations, and compare the SRH to the simulation RH. We have noted that LLPS is highly unlikely to occur for the organic acid cases based on the experiments performed by You et al. ACP 2013.

For the initial Baltimore simulations, out of 1330 simulations utilizing 1-pentanol in the non-acid organics mixture, 372 are non-LLPS (28%) while 958 (72%) are projected to experience LLPS. Based on these results, we have altered our experimental approach by replacing 1-pentanol in the analysis with a more atmospherically-relevant compound with a higher O:C ratio: 2-methyltetrol (1-methylbutane-1,2,3,4-tetrol). This significantly increases the number of model cases that are not projected to experience LLPS. In performing this new analysis, we remove points that fall into the "likely LLPS" bin according to the Bertram et al. (2011) parameterization.

In changing the analysis, we have added a section under Methods where we explicitly discuss LLPS, and the approach we take in applying the Bertram et al. (2011) parameterization. We have extended the discussion to utilize more of the explanations from Pye et al. ACP, 2018 and Zuend and Seinfeld ACP 2012 who discuss the presence of $H^+$ and water in both electrolyte- and organic-rich phases in systems that have undergone LLPS. We have added the following text (lines 346-363) to the Methods section:

"**Evaluation of LLPS and Accommodations for LLPS Scenarios**
The O:C ratio is a key factor that determines whether LLPS occurs in organic-containing particles (Song et al., 2018a; Freedman, 2017). We followed the parameterization found experimentally by Bertram et al. (2011) to evaluate the presence of LLPS in our simulations. This method uses the overall mixture O:C ratio to determine the separation RH of the mixture. If the modeled (in this case, specified/enforced) system RH is lower than the parameterized RH, LLPS is likely to occur. This was performed for each of the non-acid mixtures for both Baltimore and Beijing data to verify the claim that LLPS was not anticipated to occur. For each point in the non-acid model evaluations, the LLPS condition of Bertram et al. (2011) was employed, and the $RH_{LLPS}$ predicted by the method compared against the system $RH/H_2O$ activity reported by AIOMFAC. For cases where the parameterized $RH_{LLPS}$ was higher than the predicted system RH, LLPS was anticipated to occur, and the point was flagged and excluded from further analysis. Out of 1331 simulations, Baltimore had 55% ($n = 732$), 70% ($n = 932$), and 75% ($n = 998$) simulations that met the non-LLPS criteria at 70%, 80%, and 90% RH

respectively. Beijing had 85% (*n* = 1131), 89% (*n* = 1185), and 93% (*n* = 1238) of simulations meet the non-LLPS conditions at 70%, 80%, and 90% RH respectively. Experimental work by You et al. (2013) indicates that glutaric acid, malonic acid, oxalic acid, or their mixtures do not undergo LLPS at any of the RHs investigated."

3.) Acids vs. non-acid organics

In the abstract and similarly on page 18, line 19, it is stated that "surprisingly, non-acidic WSOC compounds were found to have a larger effect on pH than organic acids owing to their stronger impacts on γH+". It is not made clear what is surprising, please clarify. Also, how robust is this result given the choice of rather small, polar and hygroscopic organic diacids in this study. Would the same explanation hold true if a diacid of lower O:C ratio (less hygroscopic) were used? Moreover, the finding may depend on how ISORROPIA, E-AIM, UNIFAC and AIOMFAC treat organic acids (accounting for dissociation or not); see specific comments.

We have extended and enhanced this portion of the analysis by replacing 1-pentanol in the organic non-acid case with 2-methyltetrol to create a more atmospherically relevant particle composition - this has changed the results to lead to the initially hypothesize result of increasing aerosol pH with increasing organic non-acid concentration. Similarly, we have used the E-AIM model predictions of pH in the organics acids case (modeling the Beijing data in E-AIM rather than ISORROPIA) to incorporate the explicit dissociation of organic acids (as AIOMFAC was found to not include those reactions). This result is only relevant for diacids of relatively low-volatility and high solubility; for larger acids with higher O:C it is likely that the results would be equally contradictory as in the cases using 1-pentanol vs. 2-MT. The statements in the conclusion have been changed to reflect the additional analyses performed with 2-MT and the use of E-AIM pH for the organic acids cases.

**Specific Comments**

1.) Page 2, line 23: define the meaning of "aerosol strong acidity or total acidity"

The statement has been changed to include the definitions as follows: "pH, the parameter serving to define and describe the acidity of aqueous solutions, often has no direct correlation with proxy measurement methods such as aerosol strong acidity ($H^+$ contributed by strong acids that dissociated completely at any pH level) or aerosol total acidity (dissociated $H^+$ and undissociated $H^+$ bound to weak acids) (Hennigan et al., 2015;Song et al., 2018)."

2.) P3, l6: "to obtain the pH of an aerosol distribution". Since distribution is mentioned, would the same pH be expected for all aerosol particles over a considerable size range (e.g. ultrafine vs. larger accumulation mode particles)? Do the input data from measurements account for size modes?

This phrase has been changed to read "to obtain aerosol pH values." We agree with the Referee that aerosol pH does vary with particle size, but it was not our intention to include the added dimension of ambient aerosol size distributions into our analysis (not one of the aims of this investigation). For the purposes of this article, the aerosols are treated as belonging to a single aggregate aerosol phase.

3.) P3, l10 - 15: the aerosol thermodynamic models are introduced. At this point it is appropriate to define the acronyms and to cite the key references describing the models, which are missing; see associated websites. Also, citing Ganbavale et al. (2015) in the context of AIOMFAC and its web model seems inadequate as that study does not concern organic-inorganic mixtures.

The initial introduction to the models has been modified to say: "The Extended Aerosol Inorganics Model (E-AIM, http://www.aim.env.uea.ac.uk/aim/aim.php) (Wexler and Clegg, 2002;Friese and Ebel, 2010) and the ISORROPIA-II model (Greek for 'equilibrium,' http://isorropia.eas.gatech.edu) (Fountoukis and Nenes, 2007) are widely used to calculate aerosol pH for atmospheric and experimental particle distributions (Guo et al., 2017;Guo et al., 2016;Guo et al., 2015;Wang et al., 2016). The Aerosol Inorganics-Organics Mixtures Functional groups Activity Coefficient (AIOMFAC) model (http://www.aiomfac.caltech.edu/model.html) offers the most extensive treatment of organic-inorganic interactions (Zuend et al., 2008;Zuend et al., 2011) of models to date, but is primarily an activity coefficient model that does not solve full thermodynamic equilibrium calculations or phase diagrams as E-AIM and ISORROPIA do."

4.) Pages 3 and 4 (first paragraph): AIOMFAC is discussed as only being an activity coefficient model, which seems correct, but the authors forgot to mention that there are equilibrium models based on AIOMFAC, including the one used by Pye et al. (2018), Hodas et al. (2015), Zuend et al. (2010, 2012) and others. The study by Pye et al. used both ISORROPIA and AIOMFAC-based equilibrium models to estimate aerosol pH. It seems that the authors only refer to the online model of AIOMFAC, which could be made clear. Also, do ISORROPIA, E-AIM, AIOMFAC solve the ion association/dissociation equilibria, like bicarbonate and bisulfate in aqueous systems? What about organic acid dissociation? There is an E-AIM model version for specific acids, but it seems that organic acid dissociation is not considered explicitly by the models, right? Discussion of this could be important for pH calculations in the context of diacids (of different pKa values) and when contrasting the effects of acidic vs. non-acidic organics.

The version of AIOMFAC used here was the online model, used exclusively as an activity coefficient model. ISORROPIA, E-AIM, and AIOMFAC all solve the ion association/dissociation problem, especially for the bisulfate/sulfate system which is of importance. E-AIM treats explicitly the partial dissociation of species used here in both the single and double deprotonated anion forms are included in the outputs. AIOMFAC treats all organic acids as undissociated species, which informed further additions to this work detailed elsewhere.

The following clarifying statements have been added to the section to further detail the use of the models:

"While AIOMFAC has been used in combination with thermodynamic equilibrium models such as ISORROPIA-II (Pye et al., 2018), these are custom modifications to the models, and not reflected in the online versions used in this study."

"An additional consideration between the models is their treatment of organic acids. E-AIM offers support for limited (n = 8) organic acid species, and treats the dissociation equilibrium of organic acids. In contrast, AIOMFAC treats organic acids as non-dissociating, a model difference that is discussed in detail below. Note that the ion dissociation equilibria of inorganic species (such as $HSO_4^-/SO_4^{2-}$) are explicitly considered in the equilibrium calculates of all three models employed in this study."

5.) P4, l 15: Define LLPS

We have added the definition of liquid-liquid phase separation preceding the first use of the initialism LLPS.

6.) P5, l 5: Rephrase "increasing particle phase partitioning"; what kind of partitioning is meant? gas-particle, liquid-liquid?

We refer here to gas-particle phase partitioning of organic species. This comment has been clarified to say "increasing gas-particle phase partitioning."

7.) P6, l12: Rephrase "S-curves" - these are known as sigmoid curves.

We have replaced the phrase 'S-curves' with the more accurate phrase 'sigmoid curves.'

8.) P6, l16: how was the threshold uncertainty of 0.5 pH units determined? Cite a reference.

This sentence has been removed.

9.) P8, l18: What is "the standard table of AIOMFAC organic species"? this is unclear

This comment has been clarified to reflect that the organic species selected were chosen by their availability in the predefined list of organic species included in the AIOMFAC web interface.

10.) P10, l9: Clarify meaning of "Metastable mode" in this context. This seems to be a technical term used with ISORROPIA, but jargon should be avoided.

"Metastable" is not just jargon – it is used to describe a system that is not at a global equilibrium point. In this case, metastable means the system may be supersaturated with respect to the solubility of dissolved ions, but salts do not form. A simplified definition of 'metastable mode' (the formation of solids in the model being disabled) was added for clarifying purposes: "The formation of solids in the model was disabled (leading to potential supersaturated aerosols, metastable mode operation), based on the justifications in previous studies (Guo et al., 2015; 2018b)".

11.) P11, l16: "The selected species and order of allocation of the ionic species appears to be dependent solely on the researcher, and a priori knowledge of which molecular species are likely to exist in the aerosol particle as the dissociated ionic species." It is unclear what a priori knowledge is needed. Do the authors mean that the choice of species allocation affects model predictions or not?

The following clarifying statement has been added: "The selection of species is unlikely to affect model outcomes, as this is simply a way to account for the ionic species present in the AIOMFAC model inputs, which require matched cation-anion pairs, and are expected to be fully-dissociated in the aqueous phase during model evaluation."

12.) P17, l1 - 4: The statements in this sentence appear to contradict each other. Is the increase of oxalic acid from 1 to 2 μg/m3 not a discrete increase in concentration? Revise.

We have revised this statement to reflect the change from the AIOMFAC model to the E-AIM model for prediction of aerosol pH in the organic acids case. The use of E-AIM eliminates the presence of this unusual collection of points, and results in the removal of the poorly-worded section of the original manuscript.

13.) P17, l9 - 11: "Taken together, this may indicate that the system pH changes to such a point that one of the acids which may not previously have been fully dissociated begins to do so, spontaneously lowering system the pH by nearly 0.4 pH units". Are you referring to inorganic acids here or to the dicarboxylic acids? If the latter, is AIOMFAC even computing the dissociation of acids (explicitly or implicitly)?

We have addressed this comment by specifically stating that this could be the effect of organic acid dissociation. However, further insights into the working of the AIOMFAC model reveal that the model does not explicitly treat these processes. Subsequently, it may be the result of a transitional point in the $HSO_4^-/SO_4^{2-}$ equilibrium. We have accounted for this by utilizing the E-

AIM model, which does explicitly treat the acid dissociation of the selected organic acids for the prediction of aerosol pH in the organic acid cases.

14.) P18, l10: rephrase

This section has been rephrased in the following way: "The magnitudes of these observed pH changes, with the exception of the Beijing organic acids case at high ($> 25$ µg m$^{-3}$ acids concentration) is not expected to significantly alter particle conditions or lead to changes in particle chemistry, except when the pH is close to the point where a given species is almost equally partitioned between the gas and particle phases (*i.e.* on the center/vertical portion of the titration-style sigmoid curves).  This effect is demonstrated in the work of Guo et al. (2018b) and Vasilakos et al. (2018): when the pH lays on or near the inflection point of the sigmoid curve, a change of 0.5 pH units will have significant effect on species partitioning; however, when the pH is in the near-horizontal portions of the curve above or below the rapid transitional region, a change of 0.5 pH units will have negligible effect on partitioning, and thus particle chemistry."

15.) P19, l5: "the situation becomes more complicated as there exists no accepted definition of pH for a predominantly-organic phase in which the solvent is the organic aerosol constituents rather than ALW."
This referee considers this statement to be incorrect. Isn't the definition by IUPAC the accepted definition of pH, which was also used in this study? The definition of pH by IUPAC seems to be applicable in either situation; it solely depends on the H+ activity. Further, any (organic) phase containing dissolved H+ likely also contains some amount of water.

The Referee is correct and our manuscript has been corrected to state that pH could be defined in either phase.  The statement in question has been corrected to say: "As LLPS scenarios still require equilibrium between both predominantly-aqueous and predominately-organic phases, there is both water and inorganic ions (including H$^+$) in the organic phase, and organics in the inorganic-rich aqueous phase (Zuend and Seinfeld, 2012;Pye et al., 2018).  Thus the IUPAC definition of pH could be applied to either phase so long as H$^+$ activity could be defined, necessitating an understanding of if and when LLPS occurs, and the phase for which pH is being reported."

16.) P19, l10: Didn't Pye et al. (2018) consider also cases with a single aqueous phase? Their Figure 3 suggest they had both LLPS and single phase cases for the SOAS field study.

Pye et al. (2018) did consider both cases within their SOAS investigation.  Our investigation differs in that we seek only to pursue cases in which LLPS did not occur.  We have modified our analysis using the LLPS condition specified by Bertram et al. (2011) to investigate only points where LLPS did not occur.

We have modified the statement to clarify this point. The new statement reads as: "This work stands apart from, but connects to related works. Pye et al. (2018) specifically examined the effects of LLPS, but the present study examines a different particle regime altogether (single aqueous phase with water, inorganics, and organics); instances where LLPS were predicted to occur were excluded from the analysis for this reason."

17.) P21, l15: It is unclear how a "forced metastable condition" would help to avoid LLPS. These are two independent processes/states. Clarify.

This statement has been clarified to tie metastable operation to the formation of solid precipitates only. The metastable condition does not have bearing on avoiding LLPS, but rather to avoid precipitate formation, which could alter aqueous phase activity values. The statement has been clarified to read: "Because we have forced the metastable mode on our use of the models, the system mixing state becomes another potentially significant source of error. Here we have considered only internally mixed aerosol particles without LLPS, a case that may not exist given the concentration of organic species utilized in the model study; formation of solid precipitates may occur, which has the potential to drastically alter the aqueous phase activity values."

18.) P22, l20: "for especially glassy aerosols" – unclear how viscosity and glassy aerosols are related to pH in this context. Glassy aerosols would indicate significant equilibration time scale effects, but what does that have to do with pH?

This statement has been modified to remove the phrase 'glassy aerosols.' We agree with the Referee that glassy phase states would indicate significant equilibration time scales, and that in this context, the relation to pH does not make sense.

19.) P23, l4: While the authors show that the aerosol pH value is mostly determined by inorganic species, the present study and previous work cover a relatively limited set of organic compounds and conditions. Hence, general conclusions should be stated carefully. One aspect that warrants further study and discussion concerns the partitioning of organic acids that may dissociate significantly at relatively high pH (say > 4) as well as the effects of amines acting as bases similar to ammonia. Therefore, it may be premature to conclude that exclusion of organics for future aerosol pH predictions is always justified. A disclaimer in that context may be appropriate.

We have taken strides to address this point in our reanalysis of the organic acid conditions utilizing E-AIM for model prediction of pH. We have added the clarification that AIOMFAC does not explicitly treat the dissociation of organic acids, and thus in our previous analysis we were not seeing the effects of this acid dissociation. By utilizing E-AIM, our re-analysis shows that there is a more pronounced effect on aerosol pH for the Beijing case, where the pH is in the

> 4 regime. This use of E-AIM for re-analysis shows more clearly the effect different pH regimes have on the organic acid cases. However, the results for Beijing remain relatively unpronounced until unrealistic quantities of organic acids are present. We have also clarified these statements made in the Conclusions section to better represent the results of the re-analysis employed.

**Referee #2**

In "Effects of Water-soluble Organic Carbon on Aerosol pH", the authors examine changes in pH due to water solute organic acid with inorganic-organic single-phase aerosol systems, at conditions from two locations: Baltimore (more acidic) and Beijing (less acidic). In the manuscript, the authors detail use of three widely used thermodynamic models: E-AIM, ISORROPIA, and AIOMFAC for calculations of water content, activity coefficients, and pH. The results should that the impact of the organic content explored here on pH was minimal, and continues to support the role of the inorganic compounds in determining pH. However, there are some unanswered questions regarding the calculation methods, and in particular the role of dissociation equilibria of the organic acids, that should be clarified or handled in more depth. The paper would be of interest to the Atmospheric Chemistry and Physics readership, and should be considered for publications after the following are addressed.

1.) Organic acids and dissociation:
It was stated that "non-acidic WSOC were found to have a larger effect...than organic acids" due to their larger impact on H+ activity. However, it is not clear how the models used handle the concentration and the concentration, ionic strength, and pH dependent dissociation of the weak organic acids. Clarify how AIOMFAC treats the organic acids. Selection of organic acids can be put into E-AIM, both with an UNIFAC treatment, and a fitted treatment, which, I believe, treats the organic acid as non-dissociating and partially-dissociating. It would be helpful to see the impact of the two treatments in the pH results. It may be that different treatment of the dissociation equilibrium may result in more significant changes in pH by the organic acid.

The Referee makes an excellent point, and our analysis has been modified. AIOMFAC does not explicitly treat the dissociation of the organic acid species. We have switched to using E-AIM for the aerosol pH model predictions in the organic acid cases. Subsequently, we find that the organic acid dissociation has an approximately equal effect for Baltimore (where pH is low from the start), and a more obvious effect for Beijing (where pH is higher initially). By accounting for this dissociation explicitly via the E-AIM model, we observe that the hypothesized trends in pH (*e.g.* that organic acid addition should decrease pH which it is initially high, and that non-acid organics addition should generally increase pH) hold.

2.) Effects due to selected method/models:
There remains an underlying question of if the results (e.g., the minimal impact of WSOC on aerosol pH) is due to the method and models used, versus being representative of the true pH in the Baltimore and Beijing conditions. It would be helpful to compare results with various treatments of organic acid dissociation equilibrium (see above comment) as well as organic-inorganic mixtures. For the later, one can add organic compounds to E-AIM, by way of using UNIFAC functional groups – how do the results from this method compare to the that with a combined E-AIM for the inorganic, AIOMFAC for the organic method used here? E-AIM was used for the Baltimore date, and ISORROPIA for the Beijing data – how do the results vary if this was switched? For example, a model-based impact likely explains the isolated group of points in Fig 4 (page 16 line 21 – page 17 line 11) – is that same group of points found with oxalic acid is treated using a different model?

We agree with the Referee's comment, however, the suggested analysis is not possible. The online version of AIOMFAC treats organic acids as non-dissociating species. ISORROPIA does not provide inclusion of organic acids. E-AIM treats organic acids as dissociating species, able to contribute $H^+$ through full or partial dissociation. Therefore, direct comparison of predictions with the different models would be expected to be quite different. See our response to Referee #2, comment 1 above, as well.

3.) Effects of the inorganic salts and the model organics chosen:
How much impact did the choice of method to go from individual ions to salts (page 11) have on the results? How does this compare to results if the ions were evenly distributed into all the possible salt combinations? For the organics, more information on why the three particular non-acids were chosen. (line 22, page 8) Also, why were organic acids and non-organic acids not considered together? (line 5, page 9). Is it expected that having the mixture would impact the results?

We feel that the choice of salts exists primarily as an 'accounting' method to transform the outputs of the E-AIM or ISORROPIA model into inputs that AIOMFAC can interpret. As we have used measured (average) compositions, we selected salt species known to exist in inorganic aerosols; consideration of all the various salt combinations would have be prohibitive in the time frame for publication. Similarly, the exploration of organic acids and non-acid organics was not considered for similar reasons, in addition to leaving further analysis for future works once the method employed in this current study was peer-reviewed and published. However, we anticipate that having the mixtures would impact the results (*e.g.* as non-acid organics display an increase in pH placed against the downward or flat trend observed by organic acid addition).

We have addressed the Referee comments by adding the following to the discussion:

"The selection of species is unlikely to affect model outcomes, as this is simply a way to account for the ionic species present in the AIOMFAC model inputs, which require matched cation-anion pairs, and are expected to be fully-dissociated in the aqueous phase during model evaluation."

"Levoglucosan ($C_6H_{10}O_5$), tetrahydrofuran (($CH_2$)$_4$O), and 2-methyltetrol (1-methylbutane-1,2,3,4-tetrol, $C_5H_{12}O_4$), three organic species observed in ambient aerosols, were selected as the non-acid organic species."

**Minor Comments**
1.) Define what is meant by "...the thermodynamic equilibrium problem…" (line 2, page 4).

This statement was revised to explicitly state AIOMFAC is an activity coefficient model and does not solve the equilibrium calculations of E-AIM or ISORROPIA. The new line reads: "AIOMFAC offers wide support for organic components, but is an activity coefficient model that does not solve the equilibrium partitioning calculations for which the other models were designed."

2.) What are the typical organic to inorganic mass ratios of the two region (line 1, page 9)

The following statement was added to answer the Referee's question: "For Beijing, typical organic mass fractions can be on the order of 50-70% of total aerosol mass (Zhou et al., 2018), and 20-60% of total aerosol mass for continental mid-latitude locations like Baltimore (Carlton et al., 2009)."

3.) How much H+ was typically added for electroneutrality (line 19-20, page 9), compared to the other cations? Was it significant? Please comment.

[revised manuscript text omitted]